# Reveal Object in Lensless Photography via Region Gaze and Amplification

**Xiangjun Yin**[1]    **Huihui Yue**[2*]
[1]Centre for Integrated Circuits and Systems (CICS), Nanyang Technological University, Singapore
[2]School of Physical and Mathematical Sciences, Nanyang Technological University, Singapore
`xiangjun.yin@ntu.edu.sg, huihui.yue@ntu.edu.sg`

## Abstract

Detecting concealed objects, such as in vivo lesions or camouflage, requires customized imaging systems. Lensless cameras, being compact and flexible, offer a promising alternative to bulky lens systems. However, the absence of lenses leads to measurements lacking visual semantics, posing significant challenges for concealed object detection (COD). To tackle this issue, we propose a region gaze-amplification network (RGANet) for progressively exploiting concealed objects from lensless imaging measurements. Specifically, a region gaze module (RGM) is proposed to mine spatial-frequency cues informed by biological and psychological mechanisms, and a region amplifier (RA) is designed to amplify the details of object regions to enhance COD performance. Furthermore, we contribute the first relevant dataset as a benchmark to prosper the lensless imaging community. Extensive experiments demonstrate the exciting performance of our method. Our codes will be released at `https://github.com/YXJ-NTU/Lensless-COD`.

## 1 Introduction

Concealed object detection (COD) (Liu et al. (2023); Sun et al. (2024)) is an emerging task that plays an essential role in many visual applications, such as medical image analysis (Luo et al. (2022)), as it aims to extract objects hidden in the scene. Various models (Fan et al. (2022); Mei et al. (2021)) have been developed and performed well on relevant datasets collected by current imaging systems. However, owing to their bulky size, existing imaging systems cannot access the tight areas. Accordingly, developing miniature and compact imaging systems has become imperative to overcome these challenges. Due to replacing lenses with optical masks (Khan et al. (2022); Asif et al. (2017); Boominathan et al. (2020); Antipa et al. (2018)) and calculations, lensless cameras are allowed to be flexibly miniaturized (Tan et al. (2019)). Thus, the aggregation of lensless cameras into COD is a potential option.

As shown in Fig. 1 (a), unlike existing imaging systems that enable scene-resembling imaging, optical mask-based lensless camera modulates the scene radiances into encoded patterns (*i.e.*, lensless imaging measurements) without any visual information, which leads to severe challenges in COD for lensless cameras. The primary challenges are: 1) Lensless imaging lacks traditional visual features, making it challenging to extract task-relevant information from the data; 2) The complexity of the data impose greater demands on model training and optimization, particularly in noise suppression and key information retention; and 3) The inherent difficulties of the COD task itself. Existing lensless imaging studies present several closely related topics that offer valuable insights into COD within the context of lensless imaging. A recent study (Yin et al. (2022)) proposes a spatial feature learning (SFL) module for obtaining partial coarse spatial information to facilitate the detection of objects in lensless imaging while maintaining low computational costs. Inspired by this, we design an optical-aware feature extraction (OFE) module for learning spatial features. Note that unlike the supervised SFL module in (Yin et al. (2022)), our OFE module is not supervised but learns in concert with the subsequent task, thus learning more meaningful features.

Recent studies (Rao et al. (2022); Lin et al. (2023)) have shown that according to biological and psychological investigations, frequency information helps to effectively mitigate semantic clutter and

---

* Corresponding author: `huihui.yue@ntu.edu.sg`

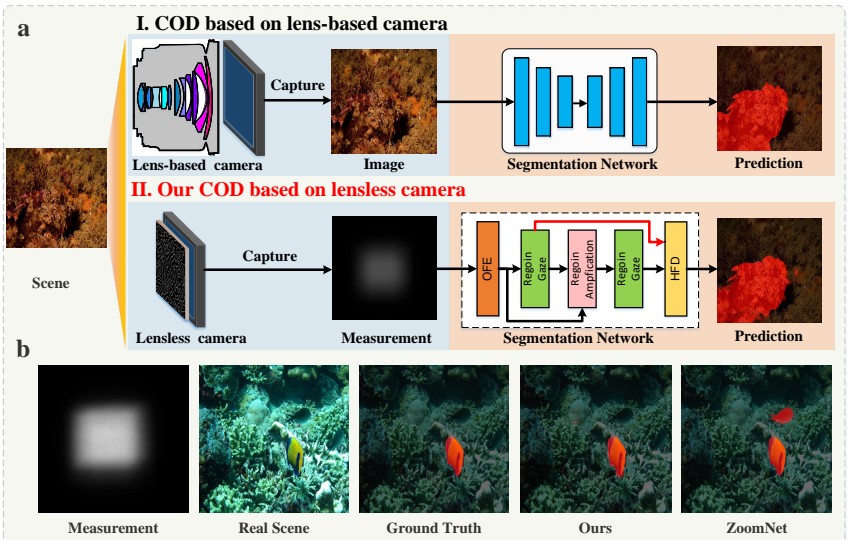

Figure 1: Illustrations of systems and visual examples of COD for lensless imaging measurements: **a**. Comparison between our COD based on lensless camera and the current COD based on traditional camera; **b**. The results by state-of-the-art COD method (ZoomNet Pang et al. (2022)) and our RGANet. Our method can mine concealed objects from the lensless imaging measurements.

improve COD performance. Therefore, to boost the detection of concealed objects against backgrounds, frequency cues are one of the key elements. Inspired by this, we propose a region gaze module (RGM), where a frequency cue encoding (FCE) component and a spatial information encoding (SIE) component are designed to extract frequency cues and spatial cues respectively, and then a spatial-frequency feature fusion (SFFF) component is built for collaborative learning of spatial cues and frequency cues. Further, another study (Xing et al. (2023)) observed that maintaining proximity to an object plays a crucial role in enhancing the perception of intricate details. This detail perception at high-level, in turn, contributes significantly to the overall process of recognizing and identifying the object. To this end, we design a region amplifier (RA) to simulate this proximity mechanism to further enhance the COD performance by locally amplifying the region details. Based on the above description, we propose a region gaze-amplification network (RGANet), which consists of an OFE module, two RGMs, and a RA, as well as a hierarchical feature decoding (HFD) module for fine reasoning. Some visualization examples in Fig. 1 demonstrate that our RGANet can effectively detect concealed objects from lensless imaging measurements. In a nutshell, our main contributions are summarized as:

- To the best of our knowledge, we are the first to investigate the detection of concealed objects in lensless imaging, demonstrating the potential of lensless imaging for various high-level tasks.

- We propose a region gaze-amplification network (RGANet) for COD in lensless imaging. In the RGANet, the region gaze modules (RGMs) based on the spatial-frequency collaborative learning strategy are proposed to recognize the object regions, and a region amplifier (RA) based on local attention is designed to amplify the region details.

- We contribute corresponding datasets as benchmarks and extensive experiments demonstrate that our method can accurately detect concealed objects from lensless imaging measurements.

## 2  RELATED WORK

### 2.1  CONCEALED OBJECT DETECTION (COD)

COD is challenging due to the high similarity between objects and their surrounding environments. Previous research has been devoted to this challenge using a variety of strategies. Le *et al.* (Trung-Nghia et al. (2019)) proposed to aggregate classifieds into pixel-level detection. Fan *et al.* (Fan et al. (2022)) developed the SINet to advance the field of COD. Chen *et al.* (Chen et al. (2022)) designed

to integrate cross-level features of concealed objects. Liu *et al.*(Liu et al. (2023)) built MSCAF-Net to focus on multi-scale context-aware cues. Mei *et al.* (Mei et al. (2021)) proposed to first localize potential objects and then focus on discovering and eliminating interferences for progressive reasoning. Liu *et al.* (Liu et al. (2022)) constructed an online confidence estimation network to model the aleatoric uncertainty for COD. Pang *et al.* (Pang et al. (2022)) used a mixed-scale triple network to focus on the objects by mimicking the zoom-in and zoom-out behavior of humans when viewing blurry images. Ma *et al.* (Ma & Sun (2023)) proposed a cross-level interaction network based on scale-aware augmentation for the COD task. However, the above studies use clear natural images as input to perform COD and cannot be directly applied to lensless imaging. By exploring spatial-frequency cues for Region Gaze and proposing local amplification mechanisms for detail magnification, we are the first to peform COD for lensless imaging measurements.

## 2.2 LENSLESS IMAGING

Lensless imaging is increasingly recognized for its potential to overcome size limitations in smartphone photography and micro-robotics applications. The core of lensless cameras is optical masks designed with various encoding elements, including amplitude masks (Khan et al. (2022); Asif et al. (2017)) and phase masks (Boominathan et al. (2020); Antipa et al. (2018)). The diverse design of optical masks has given rise to various prototypes, including the Fresnel zone aperture (FZA) camera (Wu et al. (2020; 2021)), FlatCam (Khan et al. (2022); Asif et al. (2017)), PHlatCam (Boominathan et al. (2020)), and DiffuserCam (Antipa et al. (2018); Monakhova et al. (2020); Cai et al. (2020)). They have demonstrated potential in various applications such as hyperspectral imaging Monakhova et al. (2020), fluorescence microscopy imaging (Adams et al. (2017)), light field encoding (Cai et al. (2020)), and depth information acquisition (Zheng & Salman Asif (2020)). Recently, some studies have begun to explore high-level tasks in lensless imaging, such as gender estimation (Pan et al. (2021b)), recognition (Pan et al. (2021a)), face verification (Tan et al. (2019); Cai et al. (2024)), and object segmentation (Yin et al. (2022; 2024)). These studies provide a proof-of-concept investigation for the potential of inference tasks using lensless imaging measurements. However, these above efforts have yet to demonstrate performance in performing COD task or explicitly address the grand limits posed by such methods.

## 2.3 LEARNING IN THE FREQUENCY DOMAIN

Frequency information has been widely employed in convolutional neural networks (CNNs) for image enhancement, data compression, object detection (Shao et al. (2023); Li et al. (2022); Wang & Sertel (2021); Liu et al. (2024)), *etc*. Previous studies (Wang et al. (2022)) characterized the edges of objects and smooth regions by high and low-frequency semantics, respectively. Additionally, some researchers (Mi et al. (2022)) promoted channel recognition networks into frequency domains. Compressing vision Transformers (Kong et al. (2023)) has also been explored by removing or compressing low-frequency components. Moreover, Rao *et al.* (Rao et al. (2022)) explicitly modeled the semantic information of different frequencies and accurately guided the semantic alignment of objects. In the context of COD, frequency learning has been employed to mitigate the influence of complex high-frequency texture information (Lin et al. (2023); Sun et al. (2024)). However, there still needs to be more exploration regarding the intensive prediction of the interaction between frequency and spatial domains. In contrast, our method improves COD performance by facilitating collaborative learning between the frequency and spatial domains, thereby obtaining richer cues.

## 3 METHODOLOGY

### 3.1 OVERVIEW

As depicted in Fig. 2, based on a lensless imaging prototype, *i.e.*, PHlatCam (Boominathan et al. (2020)), we propose the RGANet for COD in lensless imaging, which contains an optical-aware feature extraction (OFE) module, two region gaze modules (RGMs), a region amplifier (RA), and a hierarchical feature decoding (HFD) module. Specifically, we employ the OFE module to learn the beneficial underlying semantics for COD. Then, we propose two RGMs to progressively reason concealed objects by mining spatial and frequency cues. Furthermore, the RA embedded between the two RGMs is exploited to magnify the details of object regions to drive COD performance. Finally, the outputs of two RGMs are aggregated in the HFD module to obtain refinement results. Our RGANet provides a comprehensive workflow for performing COD on lensless imaging and demonstrates promising results.

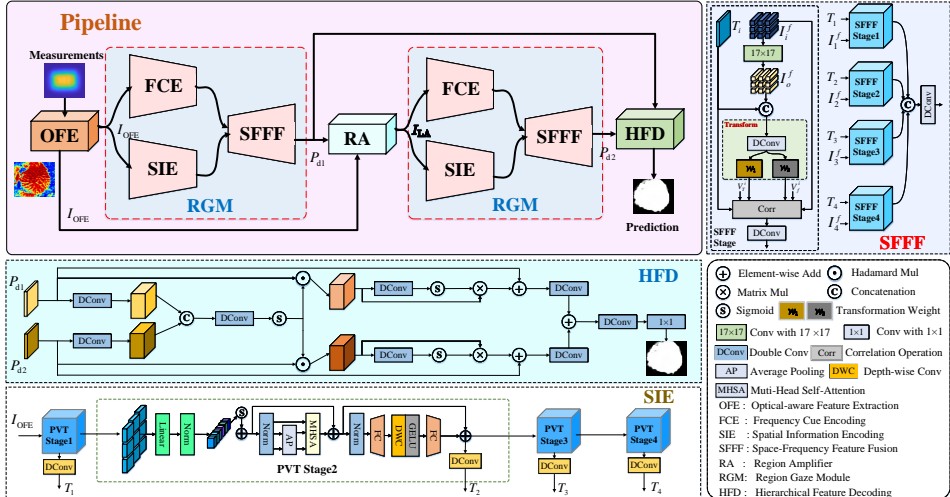

Figure 2: Overview of our RGANet. It consists of an optical-aware feature extraction (OFE) module, two region gaze modules (RGMs), a region amplifier (RA), and a hierarchical feature decoding (HFD) module. Specifically, we employ the OFE module to learn the meaningful underlying semantics for COD, and the two RGMs progressively reason concealed objects by mining spatial and frequency cues. Furthermore, the RA embedded between the two RGMs is used to magnify the details of regions to auxiliary the detection of concealed objects. Finally, the outputs of the two RGMs are fed into the HFD module to refine the final results.

## 3.2 OPTICAL-AWARE FEATURE EXTRACTION

Among various lensless imaging prototypes, PHlatCam (Boominathan et al. (2020)) is promising for its high luminous flux, lightweight, and low cost. Given this, we perform our work on PHlatCam. The imaging model of PHlatCam can be formulated as a convolution model:

$$Y = A * X + \xi, \tag{1}$$

where $*$ is the convolution operator, $A$ is the point-spread-function (PSF), which is the pattern projected onto the image sensor by the mask under the illumination of a single-point light source. $X$ is the underlying scenes, $Y$ is the measurements captured by image sensor, and $\xi$ denotes the noise.

Our work focuses on detecting concealed objects in the scene radiance $X$ from the lensless imaging measurements $Y$. However, $Y$ do not inherently contain visual cues of concealed objects. To drive COD for lensless imaging, we design an OFE module with a Wiener filtering mechanism as

$$I_{\text{OFE}} = \mathcal{F}^{-1} \left( \frac{\text{Conj}\left(\mathcal{F}(A_\theta)\right)}{K_\theta + |\mathcal{F}(A_\theta)|^2} \odot \mathcal{F}(Y) \right), \tag{2}$$

where $\mathcal{F}$ and $\mathcal{F}^{-1}$ denote the fast Fourier transform (FFT) and the inverse FFT (IFFT) operations, respectively. $\text{Conj}(\cdot)$ is the conjugate operation and $\odot$ is Hadamard multiplication. In the OFE module, PSF $A_\theta$ and regularization parameter $K_\theta$ are learnable. Note that unlike Khan et al. (2022); Boominathan et al. (2020), our OFE module does not act as visual reconstruction to satisfy the requirements of the human eye but rather collaborates with the back-end design to reason about the semantics that will benefit COD, and effectively addressing the challenges of lensless imaging.

## 3.3 REGION GAZE VIA SPATIAL-FREQUENCY COLLABORATIVE LEARNING

We propose an RGM that incorporates frequency and spatial features in a collaborative learning manner. In the RGM, a spatial information encoding (SIE) component and a frequency cue encoding (FCE) component are designed to extract spatial cues and frequency cues, respectively, and then a spatial-frequency feature fusion (SFFF) component is built for collaborative learning of spatial cues and frequency cues. In our method, two RGMs are coupled to refine the predictions incrementally. Since both RGMs share the same structure, we only detail the first RGM.

**Spatial Information Encoding.** Recent study (Wang et al. (2021)) shows that Transformer-based architectures, handling global features better than traditional CNNs. Therefore, we use the Pyramid Vision Transformer (PVTv2) to build the SIE component for extracting global features and

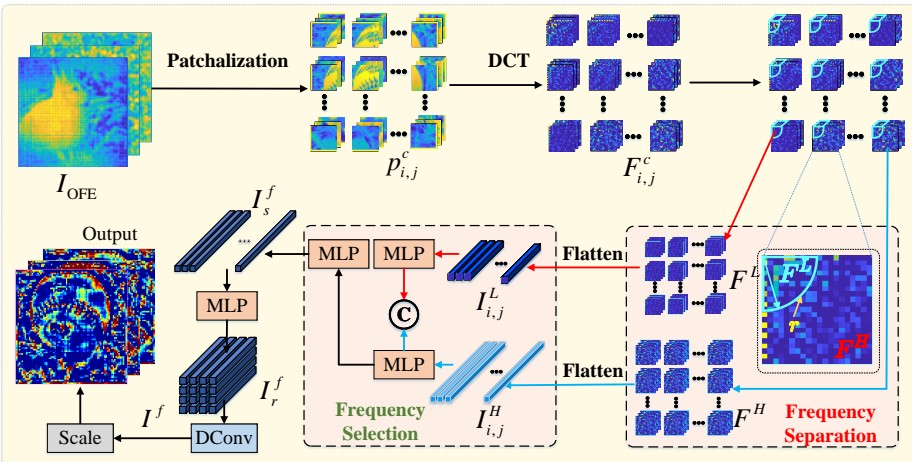

Figure 3: Illustrations of the FCE component. It contains the DCT operation, frequency separation, and frequency selection for the collection of frequency cues.

understanding lensless semantic scenes. PVTv2, with its pyramidal structure, efficiently captures multi-scale, high-resolution features through a progressive shrinking strategy and spatial reduction attention. To adapt it for the lensless COD task, we remove the classification layer and design a COD head to generate multi-scale feature maps across four stages ($T_1$, $T_2$, $T_3$, and $T_4$ from bottom to up), as illustrated in Fig. 2.

**Frequency Cue Encoding.** We design the FCE to mine frequency cues, as shown in Fig. 3.

First, we divide $I_{\text{OFE}} \in \mathbb{R}^{H \times W \times 3}$ into a set of non-overlapping $8 \times 8$ patches and obtain $\left\{ p_{i,j}^c \mid 1 \leq i \leq \frac{H}{8}, 1 \leq j \leq \frac{W}{8} \right\}$ at color channel $c$. Considering that the FFT leads to the plural that the network hardly handles, we use DCT to transform each patch into a frequency spectrum as $F_{i,j}^c = \left\{ \text{DCT}(p_{i,j}^c) \mid 1 \leq i \leq \frac{H}{8}, 1 \leq j \leq \frac{W}{8} \right\}$, $F_{i,j}^c \in \mathbb{R}^{8 \times 8}$ corresponds to 64 frequency band, each value of the feature $F_{i,j}^c$ belongs to a certain frequency band.

Second, we divide $F_{i,j}^c$ into two parts, *i.e.*, low-/high-frequency component $(F_{i,j}^L)^c$ and $(F_{i,j}^H)^c$ by

$$\begin{cases} (F_{i,j}^L)^c = F_{i,j}^c(m, n), & |m - o_m| \leq r, |n - o_n| \leq r \\ (F_{i,j}^H)^c = F_{i,j}^c(m, n), & \text{otherwise} \end{cases}, \qquad (3)$$

where $(o_m, o_n)$ denotes the coordinates of the starting point of the image, *i.e.*, $(0, 0)$. We use $F_{i,j}^c(m, n)$ to index the value of $F_{i,j}^c$ at position $(m, n)$. The $|\cdot|$ is the absolute value. Note that $r$ is a learnable parameter to achieve adaptive separation of frequencies (Each $F_{i,j}^c(m, n)$ from the same patch shares the same $r$ to maintain consistency of dimension).

Third, we flatten the spectrums into a vector that aggregates all the components with same frequency into the exact location of each channel by $I_{i,j}^H = \text{Flatten}\left((F_{i,j}^H)^c\right)$, $I_{i,j}^L = \text{Flatten}\left((F_{i,j}^L)^c\right)$.

Fourth, we adaptively select high-/low-frequency features $I_{i,j}^H$ and $I_{i,j}^L$ through some channel-mixing multi-layer perceptrons (MLP) (Tolstikhin et al. (2021)) consisting of a fully-connect (FC) layer, a batch normalization (BN), and a ReLU. Specifically, we feed $I_{i,j}^H$ and $I_{i,j}^L$ into two separate channel-mixing MLPs to filter feature and concatenate their outputs, and then use another channel-mixing MLP (from bottom to up) to adaptively select frequency features $I_s^f$. The above steps are formulated as $I_s^f = \text{MLP}\left(\text{Cat}\left(\text{MLP}\left(I_{i,j}^H\right), \text{MLP}\left(I_{i,j}^L\right)\right)\right)$, $\text{Cat}(\cdot)$ is the concatenation operation, and the $\text{MLP}(\cdot)$ is the channel-mixing MLP capturing the correlation among each patch of input features.

Finally, we obtain frequency feature $I^f$ by a channel-mixing MLP and double-layer convolution as $I^f = \text{DConv}(\text{MLP}(I_s^f))$, $\text{DConv}(\cdot)$ is two cascaded $3 \times 3$ convolution layers. To match the scale of outputs from the SIE component, $I^f$ is scaled to the corresponding size, *i.e.*, $I_1^f$, $I_2^f$, $I_3^f$, and $I_4^f$. Note that unlike the fixed thresholds used in previous studies Cong et al. (2023); Lin et al. (2023); Sun et al. (2024), here we introduce an adaptive thresholding mechanism to distinguish

high-frequency information from low-frequency components. This design advantage allows for better differentiation between noise and signal, reducing the impact of noise.

**Spatial-Frequency Feature Fusion.** The frequency features are discriminative for concealed objects, while spatial features have a larger receptive field for compensating. Therefore, we build the SFFF component to fuse the above two features, as shown in Fig. 2.

First, we filter the frequency features to extract information by $I_o^f = \mathrm{Conv}_{17 \times 17}(I_i^f)$, $i = 1, 2, 3, 4$, where $\mathrm{Conv}_{17 \times 17}(\cdot)$ is a $17 \times 17$ convolution layer.

Second, we fuse the $I_o^f$ and the features from the SIE component, and each $\{T_i, I_o^f\}$ ($i = 1, 2, 3, 4$) is concatenated and fed into two cascaded convolutions with $4\times$ channels to output $\mathcal{S}^j \in \mathbb{R}^{H \times W \times n}(j = 1, 2, 3, 4)$. We reshape $\mathcal{S}^j$ to $HW \times n$ for obtaining the fusion matrix $\mathcal{W}_1 \in \mathbb{R}^{HW \times HW}$ for space domain, and $\mathcal{W}_2 \in \mathbb{R}^{HW \times HW}$ for frequency domain by $\mathcal{W}_1 = \mathcal{S}^1 \left(\mathcal{S}^2\right)^\top$, $\mathcal{W}_2 = \mathcal{S}^3 \left(\mathcal{S}^4\right)^\top$.

Third, we align the feature maps $\{T_i, I_i^f\}$ by feature correlation operation. Specifically, the transformations $\mathcal{W}_1$ and $\mathcal{W}_2$ are multiplied with features $\{T_i, I_i^f\}$ and then multiplied with two learning vectors (*i.e.*, $V_T^i \in \mathbb{R}^{1 \times C}$ and $V_f^i \in \mathbb{R}^{1 \times C}$) respectively to adjust the channel-wise information. The output $\mathcal{I}_T^i$ and $\mathcal{I}_f^i$ are fused by addition operator. The above step is formulated as $\mathcal{I}_T^i = \mathcal{W}_1 \cdot T_i \otimes V_T^i, \mathcal{I}_f^i = \mathcal{W}_2 \cdot I_i^f \otimes V_f^i, \mathcal{I}_s^i = \mathcal{I}_T^i + \mathcal{I}_f^i$, $\otimes$ is the matrix multiplier.

Fourth, we obtain the fused features by $\mathcal{I}_{fs}^i = \mathrm{DConv}(\mathcal{I}_s^i)$. These fused features are decoded in a bottom-up manner and output the prediction maps as

$$P_\mathrm{d} = \mathrm{DConv}(\mathrm{Cat}(\mathcal{I}_{fs}^1, \mathcal{I}_{fs}^2, \mathcal{I}_{fs}^3, \mathcal{I}_{fs}^4)). \tag{4}$$

The SFFF component uses discriminative frequency information to search for concealed objects while maintaining the spatial cues to ensure the details of the entities. Note that our network uses two RGMs with outputs as shown in Eq. (4). To distinguish between the two results, we rewrite the output of the first RGM as $P_\mathrm{d1}$ and the second as $P_\mathrm{d2}$.

### 3.4 REGION AMPLIFIER VIA LOCAL ATTENTION LEARNING

Concealed objects often appear as small, obscured, or resembling the backgrounds, which makes detecting such objects susceptible to misses and errors. Inspired by the fundamental phenomenon that humans tend to magnify the difference between the observed regions and the backgrounds, we design the Region Amplifier (RA) to compress the background regions and magnify the concealed object regions. We transform the output of first RGM, *i.e.*, $P_\mathrm{d1}$, into an attention map $M$, as $M = \mathrm{Conv}_{9 \times 9}(P_\mathrm{d1})$, and the $\mathrm{Conv}_{9 \times 9}(\cdot)$ is $9 \times 9$ convolution layer, exploited to expand the originally predicted region to cover the entire object region. Then, we magnify the concealed objects based on the attention map $M$. As in (Zheng et al. (2019)), the marginal distribution is obtained by the maximization of $M$ over $x$ axis and $y$ axis as

$$M_x(n) = \sum_{s=1}^{n} \max_{1 \leq t \leq W} M_{t,s}, M_y(n) = \sum_{t=1}^{n} \max_{1 \leq s \leq H} M_{t,s}, \tag{5}$$

where $W$ and $H$ are the width and height of $M$, respectively. Given the output of OFE module, *i.e.*, $I_\mathrm{OFE}$, the sampling function $\mathcal{Q}(I_\mathrm{OFE}, M)$ is defined as

$$\mathcal{Q}(I_\mathrm{OFE}, M)_{t,s} = (I_\mathrm{OFE})_{M_x^{-1}(t), M_y^{-1}(s)}, \tag{6}$$

where $M_x^{-1}(\cdot)$ and $M_y^{-1}(\cdot)$ indicate the inverse operation of Eq. (5). Thus we can obtain the final result of local magnification as $I_\mathrm{RA} = \mathcal{Q}(I_\mathrm{OFE}, M)$. Note that our RA module aims to amplify region of interests (RoIs) for secondary recognition, which significantly enhances the reconstruction quality.

### 3.5 HIERARCHICAL FEATURE DECODING

The concealed objects are progressively highlighted by magnification and spatial-frequency cues. Since the boundaries of concealed objects are often ambiguous, the above processing performance needs to be enhanced. An intuitive method is to introduce attention mechanisms to explore the in-line association of different pixels, which may contribute to determining minor differences between

boundaries and backgrounds. As shown in Fig. 2, we propose the HFD module, which fuses the output of the two RGMs, $i.e.$, $P_{d1}$ and $P_{d2}$, and then enhances the object regions with an attention mechanism to output more refined predictions.

### 3.6 LOSS FUNCTION

To well train, we combine the weighted BCE loss $L_{wBCE}$ and weighted IoU loss $L_{wIOU}$ (Wei et al. (2020)), that is, $L_s = L_{wBCE} + L_{wIOU}$ to perform supervised learning on the outputs of the two RGMs and the final output. The final loss function is given as

$$L_{All} = L_s(P_{d1}, P_{gt}) + L_s(P_{d2}, P_{gt}) + L_s(P_{final}, P_{gt}), \tag{7}$$

where $P_{d1}$, $P_{d2}$ are the outputs of 1st RGM, 2nd RGM, respectively. $P_{final}$ is the final result of HFD module and $P_{gt}$ is the ground truth.

## 4 EXPERIMENTS

### 4.1 DATASETS

To perform COD for lensless imaging, we develop the datasets, including the training dataset and testing dataset. The dataset formation process is shown in Fig. 7 in Appendix. The specific details are as follows.

**Simulated Data Generation.** The simulated data is collected from four famous COD datasets, including CAMO (Trung-Nghia et al. (2019)), CHAMELEON (Przemysław), COD10K (Fan et al. (2022)), and NC4K (Lv et al. (2021)). We select 1857 of these images, then generate the corresponding simulated lensless imaging measurements by the following forward imaging model

$$Y^c = \mathcal{F}^{-1}\left(\mathcal{F}(A^c) \odot \mathcal{F}(X^c)\right) + \mathcal{N}(\mu^c, \sigma^c), c \in \{R, G, B\}, \tag{8}$$

where $X$ is the underlying scene, $A$ is the PSF bound to the lensless camera. The $\mathcal{N}(\mu^c, \sigma^c)$ is the noise, which we set as Gaussian distribution with $\mu^c = 0$ and $\sigma^c = 0.1 * \max(X^c)$. Then combined with the existing paired label maps, we obtain the SLCOD dataset, as shown in Fig. 7.

**Real Data Acquisition.** The real-scene data is acquired by PHlatCam from display captured dataset (Khan et al. (2022)) containing 1000 categories. First, we exclude unsuitable scenes to obtain 2600 paired images. Then we use $Eiseg$ software to annotate and acquire label maps for performing COD for lensless imaging measurements. Finally, we double-check and re-adjust the labeled maps with significant differences to keep the annotation precision, ultimately creating the DLCOD dataset as shown in Fig. 7. Note that DLCOD dataset and the dataset in Yin et al. (2022) both originate from the same source: a 10k subset of ImageNet. Unlike our dataset, Yin et al. (2022) targets general object segmentation with broader selection criteria, resulting in a larger subset of 5.9k pairs. Thus some overlap is unavoidable, our analysis shows 326 overlapping data pairs.

**Dataset Splitting.** We split the formed dataset into multiple datasets for training and testing. For training, we randomly select 2060 pairs from DLCOD and merge them with SLCOD to generate a training set containing 3917 paired data. For testing, we divide the remaining pairwise data of DLCOD into two datasets, $i.e.$, Test-Easy with 220 paired data and Test-Hard with 320 paired data, according to the difficulty of double-checking. Figure 8 shows some examples of our datasets.

### 4.2 SETUPS

**Evaluation Metrics.** We apply four evaluation metrics for comprehensive comparisons, $i.e.$, mean absolute error ($\mathcal{M}$), mean E-measure ($E_\xi$) (Fan et al. (2021)), weighted F-measure($F_\beta^w$) (Margolin et al. (2014)), and S-measure ($S_\alpha$) (Fan et al. (2017)).

**Implementation Bodies.** In our method, the backbone is initialized with the pre-trained PVT on ImageNet and subsequently trained alongside other components. The ADAM optimizer is used for training with a "cosine" learning rate scheduling policy defined as $lr = 0.5 \times init\_r \times (1 + cos(\pi * epoch/max\_epoch))$. Here, the learning rate $lr$ is initialized with $init\_r = 5 \times 10^{-4}$, and the total training epoch is set to $max\_epoch = 100$ with $epoch$ ranging from 1 to $max\_epoch$. The batch size is configured as 8. All experiments are conducted on a Linux 20.04 server with an NVIDIA GTX 3090, utilizing PyTorch 1.8.0.

### 4.3 COMPARISONS WITH THE STATE-OF-THE-ART BASELINES

**Compared Baselines.** We compare our RGANet against two kinds of baseline methods: (1) lensless inference-base methods, such as EyeCoD (You et al. (2023)), LLI_T (Pan et al. (2021a)), and

Table 1: Quantitative results of COD for lensless imaging measurements using our method and state-of-the-art baselines on Test-Easy and Test-Hard datasets. Best results are highlighted in red.

| Method | FLOPs (G) | #Param (M) | Test-Easy | | | | Test-Hard | | | |
|---|---|---|---|---|---|---|---|---|---|---|
| | | | $F_\beta^w \uparrow$ | $\mathcal{M} \downarrow$ | $E_\xi \uparrow$ | $S_\alpha \uparrow$ | $F_\beta^w \uparrow$ | $\mathcal{M} \downarrow$ | $E_\xi \uparrow$ | $S_\alpha \uparrow$ |
| EyeCoD | 84.37 | 26.92 | 0.712 | 0.131 | 0.819 | 0.791 | 0.563 | 0.162 | 0.745 | 0.710 |
| LLI_T | 44.35 | 17.23 | 0.743 | 0.110 | 0.832 | 0.802 | 0.527 | 0.167 | 0.741 | 0.651 |
| LOINet | 6.42 | 25.31 | 0.762 | 0.103 | 0.853 | 0.821 | 0.624 | 0.122 | 0.779 | 0.733 |
| MSCAF-Net | 63.04 | 30.32 | 0.697 | 0.131 | 0.812 | 0.788 | 0.563 | 0.161 | 0.790 | 0.710 |
| OCENet | 13.32 | 55.01 | 0.623 | 0.163 | 0.851 | 0.769 | 0.511 | 0.182 | 0.811 | 0.709 |
| ZoomNet | 39.41 | 32.58 | 0.714 | 0.126 | 0.821 | 0.782 | 0.619 | 0.121 | 0.804 | 0.717 |
| Ours | 48.62 | 39.45 | 0.815 | 0.079 | 0.896 | 0.834 | 0.705 | 0.098 | 0.845 | 0.770 |

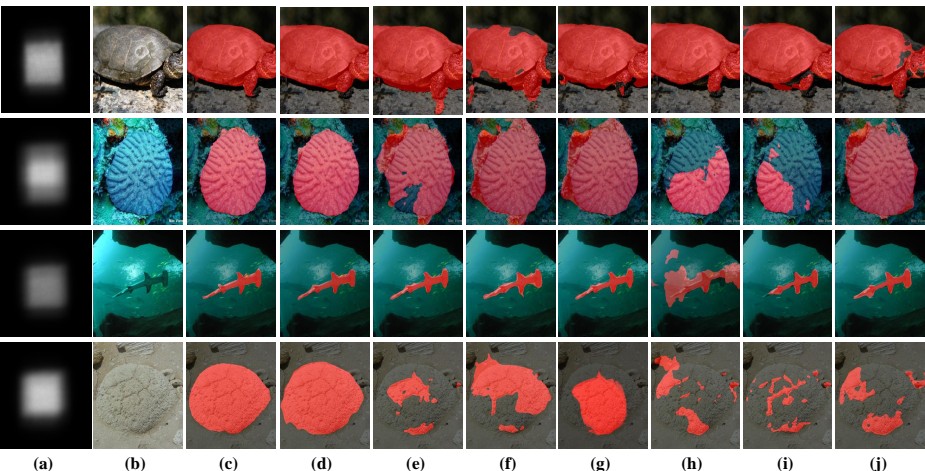

|  (a)  |  (b)  |  (c)  |  (d)  |  (e)  |  (f)  |  (g)  |  (h)  |  (i)  |  (j)  |

Figure 4: Qualitative comparisons between our method and state-of-the-art baselines on Test-Easy dataset. The (d)–(j) denote the results of RGANet (ours), EyeCoD (You et al. (2023)), LLI_T (Pan et al. (2021a)), LOINet (Yin et al. (2022)), MSCAF-Net (Liu et al. (2023)), OCENet (Liu et al. (2022)), and ZoomNet (Pang et al. (2022)). The (c) shows label maps corresponding to lensless imaging measurements (a) and underlying scenes (b).

LOINet (Yin et al. (2022)); (2) state-of-the-art COD methods, including MSCAF-Net (Liu et al. (2023)), OCENet (Liu et al. (2022)), and ZoomNet (Pang et al. (2022)). Prediction maps for all the mentioned methods are generated through re-training their models using open-source codes and a consistent OFE module for equitable comparisons. Moreover, the evaluation of all prediction maps is performed using identical code, ensuring a fair and standardized evaluation.

**Qualitative Evaluation.** Figs. 4 and 5 illustrate the qualitative results obtained on the Test-Easy and Test-Hard datasets, illustrating the effectiveness of our method in accurately performing COD for lensless imaging measurements. As demonstrated in the comparative results, our method outperforms alternative approaches in inferring a more comprehensive object structure. The success of our approach can be attributed to two key factors: (1) the incorporation of an encoder-decoder framework within the two RGMs facilitates spatial-frequency cue mining, and (2) the implementation of a local amplification mechanism contributes to the preservation of intricate details and boundaries.

**Quantitative Evaluation.** Table 1 displays the quantitative results of our method in comparison to state-of-the-art baselines on the Test-Easy and Test-Hard datasets. Our method surpasses all the compared methods across all metrics. On the Test-Easy dataset, our method achieves a significant decrease in $\mathcal{M}$ by 23.3% and an improvement in $F_\beta^w$ by 7.0% compared with the excellent method, LOINet. On the Test-Hard dataset, our method achieves a notable decrease in $\mathcal{M}$ by 19.7% and an improvement in $F_\beta^w$ by 13.0% compared with LOINet.

**Complexity Analysis.** We further provide the computational complexity of each method in terms of FLOPs and the number of parameters (#Param) in Table 1. The two metrics bound to our method are at an intermediate level among all compared methods, but our method outperforms these

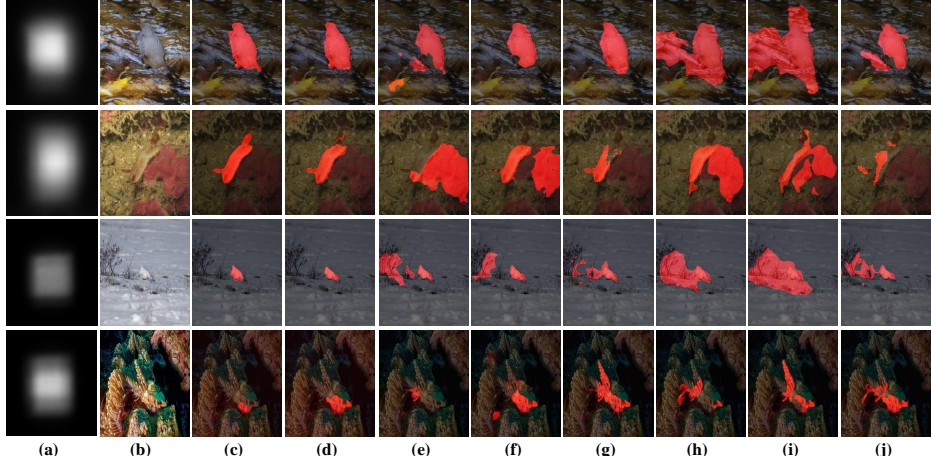

Figure 5: Qualitative comparisons between our method and state-of-the-art baselines on Test-Hard dataset. The (d)–(j) denote the results of RGANet (ours), EyeCoD (You et al. (2023)), LLI_T (Pan et al. (2021a)), LOINet (Yin et al. (2022)), MSCAF-Net (Liu et al. (2023)), OCENet (Liu et al. (2022)), and ZoomNet (Pang et al. (2022)). The (c) shows label maps corresponding to lensless imaging measurements (a) and underlying scenes (b).

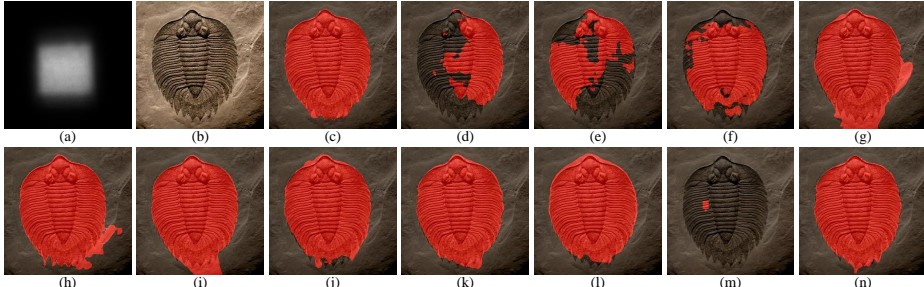

Figure 6: Qualitative comparisons of the ablation study on different configurations. The (d)–(m) denote the configurations #1 – #10, respectively. The (n) is the result with our full model (RGANet), and (c) is the label map corresponding to the lensless imaging measurement (a) and the underlying scene (b) (see Table 2 for configurations indexed by IDs).

compared methods for COD in lensless imaging. Overall, our method achieves a balance between detection performance and computational complexity.

## 4.4 ABLATION STUDIES

**The Effectiveness of RGM.** Table 2 (#4, #5, #6) reveals a significant boost in detection performance when integrating FCE into RGMs (#5, #6), relative to baseline (#4). This enhancement is corroborated by Fig. 6 (g)-(i), illustrating more complete object regions and finer details due to FCE integration. These outcomes highlight the crucial role of frequency cues in advancing the COD framework for lensless imaging. Additionally, Table 3 presents ablation studies on frequency adaptive selection within FCE, emphasizing the significance of balanced low- and high-frequency utilization. An optimal ratio ($r = 5$) consistently outperforms other configurations ($r = 1$, $r = 7$, and exclusions), underscoring the necessity of frequency adaptation for effective RGANet application in COD. Table 2 (#1, #7, #11) shows that reducing any RGM degrades performance, emphasizing their role in ensuring precision and reliability. Visual comparisons in Fig. 6 (d) and (j) further confirm that fewer RGMs result in lower performance and visual quality.

**The Effectiveness of RA.** The evaluation results presented in Table 2 (#9, Ours) highlight the significant enhancement in COD performance achieved through the incorporation of the RA. Conversely, the exclusion of RA results in a substantial decline in performance. Furthermore, as depicted in Fig. 6 (l) and (n), RA effectively enhances local details, illustrating its ability to amplify concealed object features. This observation validates RA's capacity to focus on local intricacies, allowing the

Table 2: Quantitative comparisons of the ablation study on different configurations. Best results are highlighted in red.

| ID | OFE | 1-st RGM | | | RA | 2-nd RGM | | | HFD | Test-Easy | | Test-Hard | |
|---|---|---|---|---|---|---|---|---|---|---|---|---|---|
| | | FCE | SIE | SFFF | | FCE | SIE | SFFF | | $F_\beta^w \uparrow$ | $\mathcal{M} \downarrow$ | $F_\beta^w \uparrow$ | $\mathcal{M} \downarrow$ |
| #1 | ✓ | ✓ | ✓ | ✓ | | | | | | 0.509 | 0.259 | 0.468 | 0.306 |
| #2 | ✓ | | ✓ | ✓ | | | ✓ | ✓ | | 0.624 | 0.163 | 0.511 | 0.182 |
| #3 | ✓ | | ✓ | ✓ | ✓ | | ✓ | ✓ | | 0.651 | 0.139 | 0.557 | 0.158 |
| #4 | ✓ | | ✓ | ✓ | ✓ | | ✓ | ✓ | ✓ | 0.682 | 0.134 | 0.564 | 0.153 |
| #5 | ✓ | ✓ | ✓ | ✓ | ✓ | | ✓ | ✓ | ✓ | 0.721 | 0.122 | 0.596 | 0.147 |
| #6 | ✓ | | ✓ | ✓ | ✓ | ✓ | ✓ | ✓ | ✓ | 0.718 | 0.125 | 0.592 | 0.145 |
| #7 | ✓ | ✓ | ✓ | ✓ | | ✓ | ✓ | ✓ | | 0.729 | 0.116 | 0.601 | 0.142 |
| #8 | ✓ | ✓ | ✓ | ✓ | ✓ | ✓ | ✓ | ✓ | | 0.795 | 0.087 | 0.672 | 0.116 |
| #9 | ✓ | ✓ | ✓ | ✓ | | ✓ | ✓ | ✓ | ✓ | 0.756 | 0.106 | 0.617 | 0.134 |
| #10 | | ✓ | ✓ | ✓ | ✓ | ✓ | ✓ | ✓ | ✓ | 0.423 | 0.382 | 0.392 | 0.361 |
| #11 | ✓ | | | | ✓ | ✓ | ✓ | ✓ | ✓ | 0.631 | 0.157 | 0.539 | 0.162 |
| Ours | ✓ | ✓ | ✓ | ✓ | ✓ | ✓ | ✓ | ✓ | ✓ | 0.815 | 0.079 | 0.705 | 0.098 |

Table 3: Quantitative ablation analysis of frequency adaptive selection in FCE. Best results are highlighted in red.

| ID | Configuration | Test-Easy | | Test-Hard | |
|---|---|---|---|---|---|
| | | $F_\beta^w \uparrow$ | $\mathcal{M} \downarrow$ | $F_\beta^w \uparrow$ | $\mathcal{M} \downarrow$ |
| #11 | w/o. $F^H$ | 0.732 | 0.109 | 0.608 | 0.139 |
| #12 | w/o. $F^L$ | 0.769 | 0.102 | 0.656 | 0.122 |
| #13 | $r = 1$ | 0.774 | 0.098 | 0.662 | 0.114 |
| #14 | $r = 7$ | 0.782 | 0.091 | 0.675 | 0.103 |
| Ours | Full model ($r = 5$) | 0.815 | 0.079 | 0.705 | 0.098 |

network to extract more valuable information from specific regions and contributing to the overall improvement in COD performance.

**The Effectiveness of HFD Module.** The evaluation results presented in Table 2 (#8, Ours) underscore the superior performance of our proposed HFD module in terms of COD accuracy. The visualizations in Fig. 6 (k) and (n) provide additional insights, illustrating that the HFD module effectively integrates information from the two RGMs to compensate for missing features. This integration leads to more refined and visually enhanced results, highlighting the capacity of the HFD module to improve the overall effects of COD.

**The Effectiveness of OFE Module.** The results presented in Table 2 (#10, Ours) underscore the pivotal role of the OFE module in COD for lensless imaging. Specifically, the removal of the OFE module leads to a decline in COD performance, while its inclusion enhances COD accuracy. Additionally, a visual comparison in Fig. 6 (m) and (n) further supports these findings, emphasizing that the absence of the OFE module hampers practical performance, while its inclusion enables accurate and meaningful generalization of results. This observation suggests that incorporating the OFE module provides a rich source of underlying semantics that significantly benefits the COD task.

## 5 CONCLUSION

This paper aims to solve the challenge of accurate COD on lensless imaging measurements by proposing a region gaze-amplification network (RGANet). In the RGANet, we incorporate frequency and spatial cues by RGMs for driving reasoning performance and introduce the RA embedded in the two RGMs to magnify potential concealed objects for refining final results. Furthermore, we present the first dataset for COD on lensless imaging measurements, which serves as a benchmark for future research. Experimental results demonstrate the superiority of the RGANet over existing methods and highlight its potential in advancing the community of lensless imaging.

## 6 Ethics Statements

This study advances lensless imaging for concealed object detection (COD) with potential applications in medical imaging, security screening, and camouflage identification. We ensure responsible research practices by adhering to ethical considerations. Our work does not involve personal or sensitive data, and the contributed dataset is synthetically generated or collected under controlled conditions to avoid privacy concerns. While the method may benefit medical and security applications, it is intended purely for scientific advancement, and any deployment in sensitive areas should comply with ethical guidelines and regulatory approvals. We also strive to mitigate biases in our dataset and model to ensure fairness and generalizability across imaging conditions. By following these principles, we aim to contribute positively to computational imaging and COD research while ensuring ethical and responsible development.

## 7 Acknowledgement

We would like to acknowledge Centre for Integrated Circuits and Systems (CICS) and School of Physical and Mathematical Sciences (SPMS), Nanyang Technological University (NTU), Singapore for their support in this research. Additionally, we appreciate Jingyu Yang, Kun Li, and Huanjing Yue from Tianjin University, China for their insightful discussions during the experiments, which provided valuable guidance for the successful progress of our work.

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

## A  APPENDIX

### A.1  DATASET

To reduce data costs, we combined simulated and real data to build the dataset for model training. The data generation process is shown in Fig. 7. The collection of real data requires adherence to the following experimental configuration:

(1) Distance between the PHlatCam and the display: The PHlatCam was positioned 42 cm from the display throughout all real-capture experiments. This distance was carefully selected to optimize image capture, considering the camera's field of view and resolution. This configuration remained consistent during both training and testing phases, ensuring uniform alignment of camera and monitor pixels.

(2) Display specifications:

Model and Type: The display used was a Dell S2425HS, which is an LCD screen.

Size: The screen size was 24 inches, with a resolution of 1920×1080 pixels.

Additional Notes: The image was resized via bicubic interpolation to fit the largest central square on the monitor. The white balance for PHlatCam was calibrated using the automatic white balance setting of the PointGrey Flea3 camera, determined when an all-white image was displayed on the monitor. The exposure time was governed by automatic mode of camera, with gain fixed at 0 dB.

Data collection is completed based on the aforementioned processing. Based on these, we divided the collected datasets into separate sets for training and testing. For training, we randomly selected 2060 pairs from DLCOD and combined them with SLCOD, creating a training set with 3917 pairs. For testing, the remaining DLCOD data was split into two sets based on verification difficulty: Test-Easy with 220 pairs and Test-Hard with 320 pairs. Figure 8 presents examples from these datasets.

### A.2  COMPARISONS WITH DETECTION-AFTER-RECONSTRUCTION STRATEGY

To investigate the effect of the detection-after-reconstruction strategy, we employ the state-of-the-art FlatNet (Khan et al. (2022)) for reconstruction. Subsequently, we apply EyeCoD (You et al. (2023)), LLI_T (Pan et al. (2021a)), LOINet (Yin et al. (2022)), MSCAF-Net (Liu et al. (2023)), OCENet (Liu et al. (2022)), and ZoomNet (Pang et al. (2022)) for COD. For equitable comparisons, our method replaces the OFE module with FlatNet. In Fig. 9 (b), the reconstructed image by FlatNet closely resembles the underlying scene in Fig. 9 (a).

When performing COD based on the reconstructed results, our method (Fig. 9 (d)) consistently outperforms advanced COD methods such as MSCAF-Net(Liu et al. (2023)) (Fig. 9 (h)), OCENet(Liu

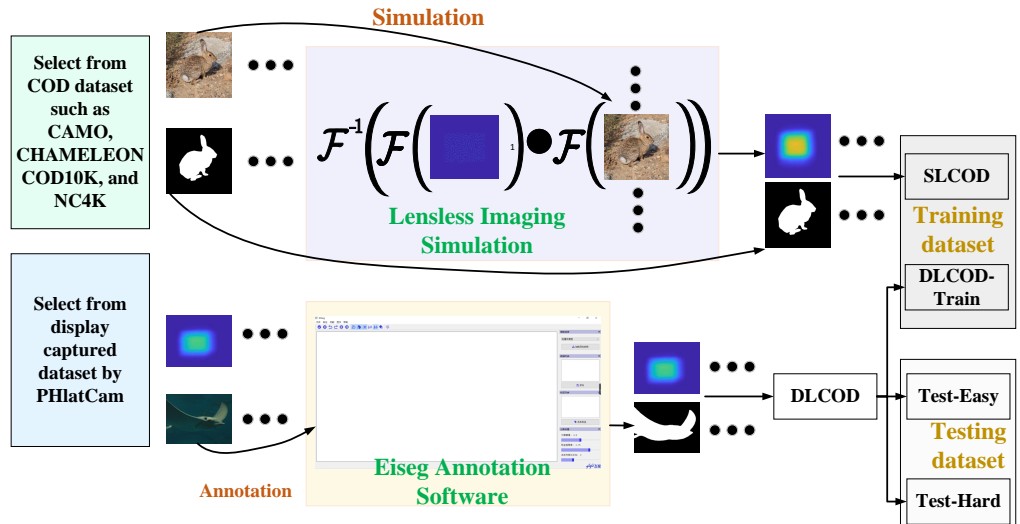

Figure 7: The formation process of our datasets for performing COD of lensless imaging measurements.

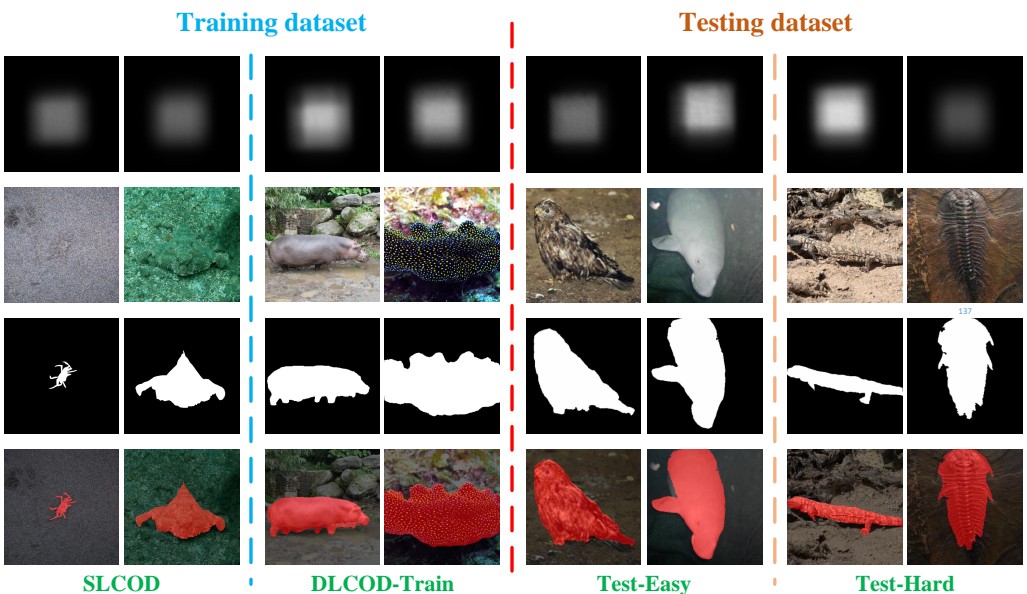

Figure 8: Examples from our datasets. Each row, from top to bottom, represents lensless imaging measurements, underlying scenes, label maps, and annotation visualizations (*i.e.*, label maps overlaid on underlying scenes for annotation clarity).

et al. (2022)) (Fig. 9 (i)), and ZoomNet(Pang et al. (2022)) (Fig.9 (j)). Additionally, we provide quantitative results of the detection-after-reconstruction strategy in Table 4. Compared with the results in Table 1, the detection-after-reconstruction paradigm yields improvements of up to $10\%$ for all compared methods in terms of $F_\beta^w$, at the cost of huge computational complexity. These findings validate the promising potential of direct COD on lensless imaging measurements.

## A.3 COMPARISONS WITH LATEST COD METHODS

To further validate the superiority of our method, we select two recent methods (FEDER He et al. (2023) and FSPNet Huang et al. (2023)) and conducted comparative experiments based on the settings and open-source code provided earlier. The corresponding qualitative and quantitative results

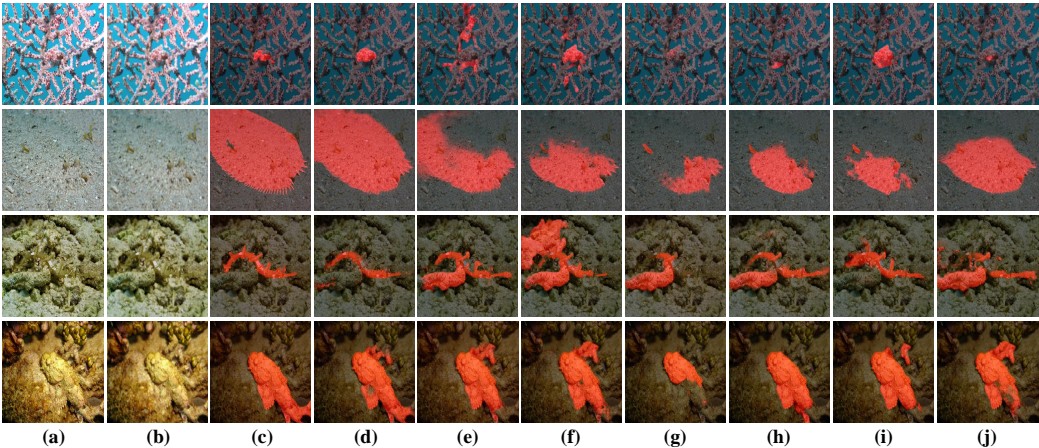

|  |  |  |  |  |  |  |  |  |  |
|---|---|---|---|---|---|---|---|---|---|
| **(a)** | **(b)** | **(c)** | **(d)** | **(e)** | **(f)** | **(g)** | **(h)** | **(i)** | **(j)** |

Figure 9: Qualitative comparisons of COD using the detection-after-reconstruction strategy on Test-Hard dataset. The (d)–(j) denote the results of RGANet (ours), EyeCoD (You et al. (2023)), LLI_T (Pan et al. (2021a)), LOINet (Yin et al. (2022)), MSCAF-Net (Liu et al. (2023)), OCENet (Liu et al. (2022)), and ZoomNet (Pang et al. (2022)). The (a) depicts underlying scenes, (b) shows reconstructions by FlatNet (Khan et al. (2022)), and (c) illustrates label maps.

Table 4: Quantitative results of COD for lensless imaging measurements using the detection-after-reconstruction strategy on Test-Easy and Test-Hard datasets.

| Method | FLOPs (G) | #Param (M) | Test-Easy | | | | Test-Hard | | | |
|---|---|---|---|---|---|---|---|---|---|---|
|  |  |  | $F_\beta^w \uparrow$ | $\mathcal{M} \downarrow$ | $E_\xi \uparrow$ | $S_\alpha \uparrow$ | $F_\beta^w \uparrow$ | $\mathcal{M} \downarrow$ | $E_\xi \uparrow$ | $S_\alpha \uparrow$ |
| FlatNet + EyeCoD | 204.27 | 86.32 | 0.810 | 0.085 | 0.823 | 0.807 | 0.708 | 0.091 | 0.832 | 0.794 |
| FlatNet + LLI_T | 164.25 | 76.63 | 0.836 | 0.063 | 0.887 | 0.859 | 0.729 | 0.075 | 0.847 | 0.835 |
| FlatNet + LOINet | 126.32 | 84.71 | 0.843 | 0.054 | 0.897 | 0.868 | 0.751 | 0.063 | 0.866 | 0.827 |
| FlatNet + MSCAF-Net | 182.94 | 89.72 | 0.831 | 0.071 | 0.889 | 0.851 | 0.737 | 0.078 | 0.856 | 0.804 |
| FlatNet + OCENet | 133.22 | 114.41 | 0.829 | 0.057 | 0.876 | 0.853 | 0.741 | 0.071 | 0.852 | 0.816 |
| FlatNet + ZoomNet | 159.31 | 91.98 | 0.847 | 0.051 | 0.903 | 0.871 | 0.752 | 0.059 | 0.869 | 0.831 |
| FlatNet + Ours | 48.62 | 39.45 | 0.815 | 0.079 | 0.896 | 0.834 | 0.705 | 0.098 | 0.845 | 0.770 |

are presented in Fig. 10 and Table 5. As evidenced by the results, the comparison methods show significantly lower performance in terms of object detection completeness, with the quantitative metrics further supporting this conclusion.

## A.4 COMPARISONS WITH STATE-OF-THE-ART COD METHODS ON REAL NATURAL SCENE

To verify the generalization ability in natural scenes, we conduct additional experiments using a dataset that captures concealed scenarios in natural environments, free from screen-based biases. This dataset, consisting of 34 pairs of lensless imaging data, reconstructed scenes, and ground truths, covers a broader range of wavelengths, allowing us to evaluate the model performance in unfiltered, real-world conditions. The results shown in Fig. 11 highlight the effectiveness and robustness of our method in real-world conditions, extending its capabilities beyond screen-based data.

Table 5: Quantitative results by latest COD methods. Best results are highlighted in red.

| Method | Test-Easy | | | | Test-Hard | | | |
|---|---|---|---|---|---|---|---|---|
|  | $F_\beta^w \uparrow$ | $\mathcal{M} \downarrow$ | $E_\xi \uparrow$ | $S_\alpha \uparrow$ | $F_\beta^w \uparrow$ | $\mathcal{M} \downarrow$ | $E_\xi \uparrow$ | $S_\alpha \uparrow$ |
| FEDER | 0.741 | 0.113 | 0.837 | 0.797 | 0.539 | 0.159 | 0.746 | 0.712 |
| FSPNet | 0.757 | 0.105 | 0.845 | 0.816 | 0.584 | 0.131 | 0.757 | 0.724 |
| Ours | 0.815 | 0.079 | 0.896 | 0.834 | 0.705 | 0.098 | 0.845 | 0.770 |

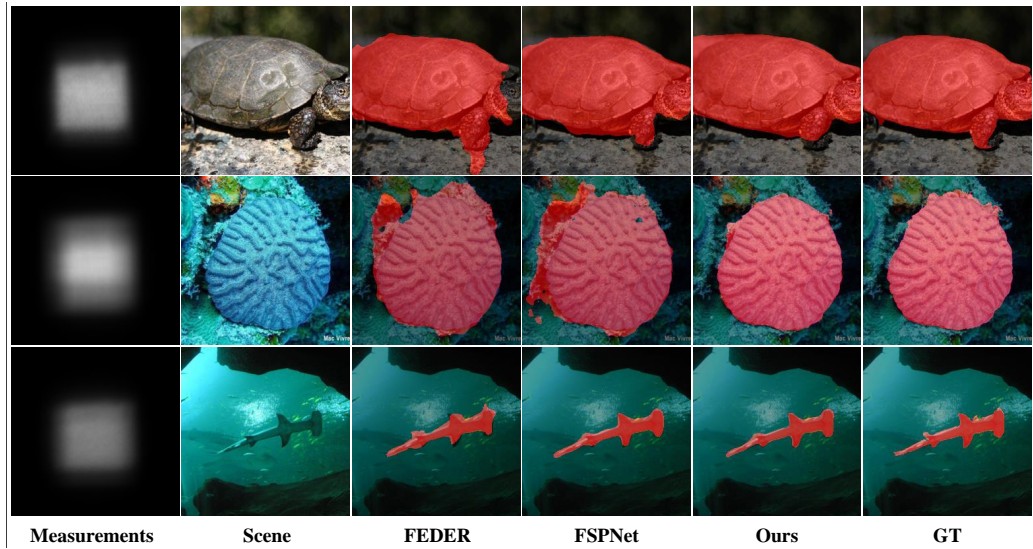

| Measurements | Scene | FEDER | FSPNet | Ours | GT |
|---|---|---|---|---|---|

Figure 10: Qualitative results with the latest COD methods, such as FEDER He et al. (2023), FSP-Net Huang et al. (2023), and ours.

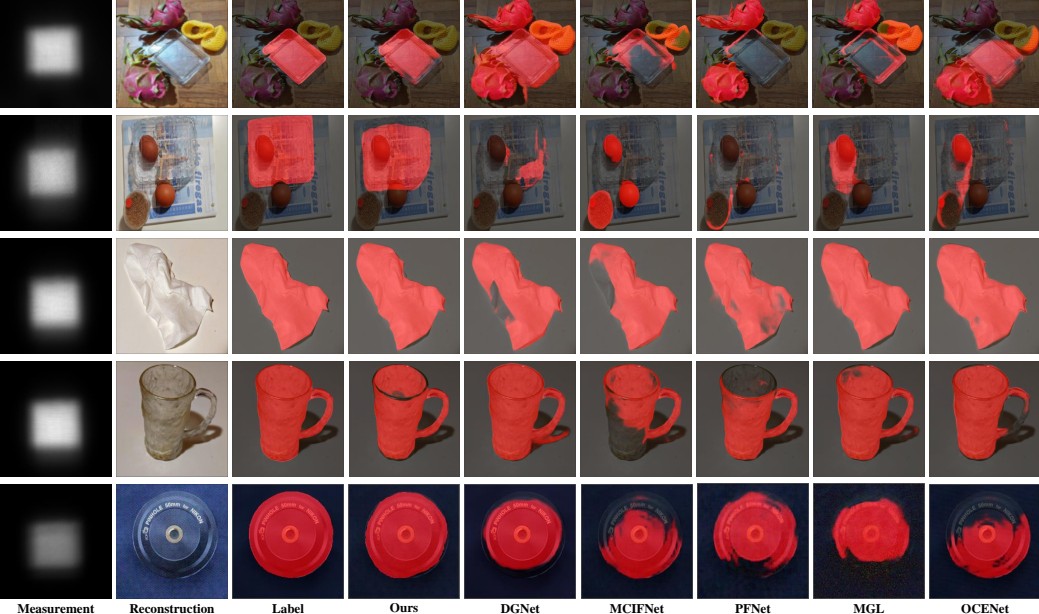

| Measurement | Reconstruction | Label | Ours | DGNet | MCIFNet | PFNet | MGL | OCENet |
|---|---|---|---|---|---|---|---|---|

Figure 11: Qualitative comparisons between our method and state-of-the-art COD methods on real natural scenes.

## A.5 DISCUSSIONS AND LIMITATIONS

Our method detects concealed objects for lensless imaging with high precision as illustrated in the above results. However, our method struggles to reason well in scenes with low contrast, extremely cluttered backgrounds and small objects, as illustrated in Fig. 12. Specifically, concealed objects implied in the above scenes suffer from incomplete detection and incorrect object location by our method and advanced methods, such as EyeCoD and ZoomNet. In the future, we will dedicate more efforts to improving the COD accuracy in these cases.

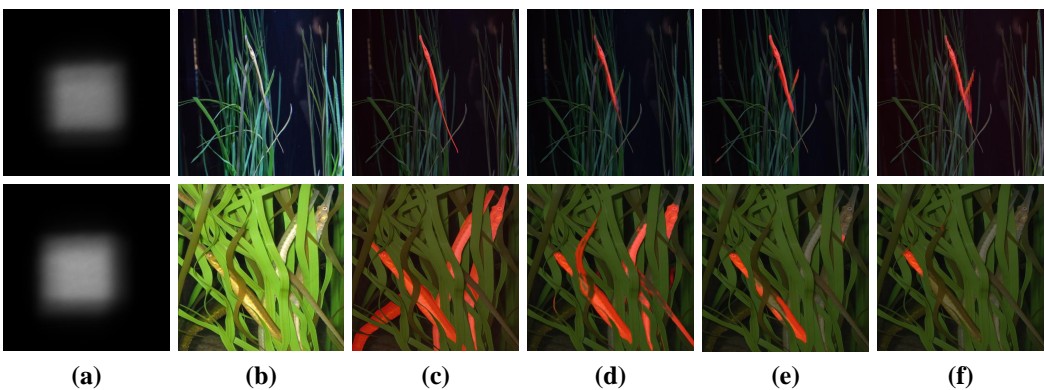

|  (a)  |  (b)  |  (c)  |  (d)  |  (e)  |  (f)  |

Figure 12: Illustration of failure cases. Examples are presented from (d) our method and state-of-the-art methods, such as (e) EyeCoD and (f) ZoomNet. The (a)–(c) denote lensless imaging measurements, underlying scenes, and label maps, respectively.

