# OpenReview forum: "Reveal Object in Lensless Photography via Region Gaze and Amplification"
_ICLR.cc/2025/Conference — ICLR 2025 Poster_

### Official Review · Reviewer_ZAok · 2024-10-29

**Soundness:** 3
**Presentation:** 3
**Contribution:** 3
**Rating:** 6
**Confidence:** 4

**Summary:**

The authors propose a region gazeamplification network (RGANet) for progressively exploiting concealed objects from lensless imaging measurements.

**Strengths:**

1. PHlatCam dataset is semantically labeled and contributes to the corresponding dataset as a benchmark and extensive experiments.
2. investigate the detection of concealed objects in lensless imaging scenarios.

**Weaknesses:**

1. The use of RGM twice in the network is confusing. The ablation experiment only studies the effectiveness of the combination of internal modules under the setting of RGM twice. It sounds more reasonable to input RGM after I_OFE passes through RA.
2. Under the condition of lensless cameras, the author studies concealed object detection. Such a combination of tasks makes people doubt whether its real application scenarios are wide. Why not study more common object detection tasks?
3. The design of the entire network framework and internal modules is relatively ordinary. Basically, it is based on existing network modules with certain modifications, giving people a feeling of an A+B combination.
4. The format and layout are uncomfortable, for example, the formulas in the paper have larger line spacing. In addition, the figures in the paper are not beautiful enough, and the color matching is abrupt.
5. From the ablation experiment result #10 in Table 2, we can see that OFE is the most important core part of the network. I think OFE+encoder-decoder can achieve good results.

**Questions:**

See Weaknesses for details.

---

> ### Author Response · Authors · 2024-11-25
>
> We sincerely thank you for your valuable comments and recognition, and have provided individual responses to address each concern clearly and thoroughly.
>
> **Weaknesses:**
>
> **Q1:The use of RGM twice in the network is confusing.**
>
> Thank you for your feedback. The lensless-based COD task is inherently challenging due to the difficulties of vision-independent imaging in lensless systems and the complexities of COD itself, which conventional methods often struggle to address. Our method, however, improves lensless-based COD accuracy by utilizing two RGMs in a structured and purposeful manner. The first RGM performs coarse extraction to identify the general region of the object, providing an initial understanding of the scene. The Region Amplifier (RA) then highlights and amplifies the output from the Optical-aware Feature Extractor (OFE). The second RGM further refines this enhanced output, and the final fusion, assisted by Hierarchical Feature Decoding (HFD), combines the results of the two RGMs to extract complementary information, optimizing COD performance. Experimental results demonstrate that our method significantly outperforms others, validating its effectiveness. To further support this, as your suggestion, we have included ablation results in the **Tab. 2** of revised manuscript. Here, in following Tabs. 1 and 2, we briefly show the results that remove the first RGM (i.e. configuration to input RGM after I_OFE passes through RA). These results show a clear decline in performance, further supporting the rationale behind incorporating both RGMs.
>
> **Table 1** Ablation study on RGM onTest-Easy Dataset
>
> | ID   | Configuration            |$F_{\beta}^{\omega}$ | $M$ |
> |------|--------------------------|------------------------|-------|
> | #1   | Full model (w/o 1-st RGM) | 0.631                  | 0.157 |
> | #2   | Full model                | 0.815                  | 0.079 |
> |     |
>
> **Table 2** Ablation study on RGM onTest-Hard Dataset
>
> | ID   | Configuration            | $F_{\beta}^{\omega}$ | $M$ |
> |------|--------------------------|------------------------|-------|
> | #1   | Full model (w/o 1-st RGM) | 0.539                  | 0.162 |
> | #2   | Full model                | 0.705                  | 0.098 |
> |      |
>
>
> **Q2:Why not study more common object detection tasks?**
>
> We greatly appreciate your concerns regarding the task we have conducted. This work builds upon our prior research on lensless-based common object detection, serving as an advanced extension tailored to more challenging scenarios. While it remains applicable to simpler tasks like common object detection, its primary focus is to address complex, confined environments. Examples include in-body detection with lensless endoscopy, where space is limited and visibility is challenging, or robotics navigating concealed and cluttered terrains that demand precise detection for effective navigation and task execution. By tackling these challenges, we aim to further the practical applications of lensless imaging technology and unlock its potential in specialized domains.
>
> **Q3:The design of the entire network framework and internal modules is relatively ordinary.**
>
> As previously mentioned, lensless-based COD tasks are inherently challenging, and a single network architecture cannot effectively address these tasks. To overcome this, we propose the Region Gaze, Amplification, and Gaze-Again (RGA) mechanism, which progressively refines detection results through a multi-stage method. The OFE is tailored specifically to extract features critical for COD tasks, while the RGM introduces a learnable threshold to adaptively separate high- and low-frequency information, overcoming the limitations of fixed-frequency methods common in existing methodes. Additionally, our RA module uniquely amplifies object regions to support fine-grained detection. Together, these components establish a comprehensive paradigm for lensless-based COD tasks, transcending individual module designs to deliver a system optimized for the unique demands of lensless imaging. This framework aims to advance the practical applications of lensless imaging technology. Therefore, our framework is carefully structured and thoughtfully designed by based on the RGA mechanism, rather than being a mere stack or assembly of modules.
>
> **Q4:The format and layout are uncomfortable.**
>
> Thank you for your insightful comments regarding the format and layout of our manuscript. In light of your suggestions, we implement a unified revision of the figures, tables, and equations in the updated version to improve their overall clarity and aesthetic presentation.

---

> > ### Author Response · Authors · 2024-11-25
> >
> > **Q5: I think OFE+encoder-decoder can achieve good results.**
> >
> > We sincerely appreciate your constructive feedback. In our comparative experiments, all the methods used for comparison were integrated with the OFE to ensure a fair evaluation, as detailed in “Sec 4.3 Compared Baselines”. These setups of comparison methods can all be described as OFE combined with a specific encoder-decoder architecture. However, the results in **Figs. 4-5 and Tab.1** clearly indicate that these comparison methods do not surpass the performance of our proposed method.

---

> > > ### Author Response · Authors · 2024-12-03
> > >
> > > Given our thorough response, we sincerely hope that you find it satisfactory and that it effectively addresses your concerns. We would greatly appreciate your feedback on whether our clarifications have resolved the issues raised.

---

### Official Review · Reviewer_EKVY · 2024-11-03

**Soundness:** 2
**Presentation:** 3
**Contribution:** 2
**Rating:** 6
**Confidence:** 4

**Summary:**

The authors aim to address concealed object detection (COD) with lensless imaging systems. They propose a network for COD leveraging spatial and frequency feature enhancement and fusion. They also annotate 2600 paired images from the Display Captured Dataset to build a new dataset for COD with lensless imaging systems. However, the authors should design a special network for this task by considering the characteristics of the lensless camera, and clarify the details of the proposed dataset.

**Strengths:**

1. A new dataset for COD with lenless imaging system.
2. Good performance for a new setting.

**Weaknesses:**

1. Straightforward method with limited novelty. The authors do not analyze the challenge for COD with lensless cameras. The main difference in design for the lensless cameras is the optical-aware feature extraction, but it refers to [1]. In addition, the main module of the proposed method is the spatial-frequency enhance module, which directly uses the idea of existing works for COD [2, 3].

2. Insufficient experiments. The authors should compare with sota COD methods, such as [2,3,4]. Moreover, they should compare with the two-stage methods (Lensless imaging methods combined with COD methods). With the large parameters and computational cost, the two-stage methods may be more lightweight.

3. Details of the proposed dataset. The authors should provide a detailed analysis of the proposed dataset to clarify the difference between the dataset from [5], since some samples shown in the paper are the same as samples in [5].

[1] PhlatCam: Designed Phase-Mask Based Thin Lensless Camera. TPAMI2020.
[2] Frequency perception network for camouflaged object detection. MM2023.
[3] Camouflaged object detection with feature decomposition and edge reconstruction. CVPR2023.
[4] Feature shrinkage pyramid for camouflaged object detection with transformers.CVPR2023.
[5] Inferring Objects FromLensless Imaging Measurements. TCI2022

**Questions:**

1. Does the performance improvement come from the large parameters and computational cost?
2. Error in Figure 2, where the sigmoid function is directly fed to addition without any input in PVT.

---

> ### Author Response · Authors · 2024-11-25
>
> We greatly value your profound insights and acknowledgment, and have provided meticulous, point-by-point responses to address each concern with precision and clarity.
>
> **Q1：About Challenge of COD with Lensless Cameras and Contributions**
>
> - **Challenge of COD with Lensless Cameras:** Currently, the challenges we face in our work primarily stem from the inherent difficulties of lensless imaging and the complexity of the COD task itself. The primary challenges are: 1) Lensless imaging lacks traditional visual features, making it challenging to extract task-relevant information from the data; 2) The complexity of the data impose greater demands on model training and optimization, particularly in noise suppression and key information retention; And 3) the inherent challenges of the COD task itself. We have revised and clarified this section in the updated version of the “Introduction”.
> - **Novel Contributions:** Our contributions should be highlighted as follows: This work is the first to address COD specifically tailored for lensless imaging, thereby extending the practical application of lensless imaging technology to inference tasks.
>   For optical-aware feature extraction (OFE), while [1] employs Wiener filter principles for reconstruction tasks, our method fundamentally diverges in its application. Unlike [1], which optimizes for visually improved reconstructed images, our OFE is embedded in a framework jointly trained for COD. This design ensures the extracted features are specialized for COD, aligning with task-specific requirements rather than general-purpose visual fidelity. Moreover, our method leverages these tailored features to enhance COD performance through end-to-end optimization. This method represents a critical advancement in overcoming the unique challenges of lensless imaging, including complex scenes and unconventional data characteristics, which traditional methods struggle to address effectively. We have clarified this distinction in the revised manuscript to better highlight the unique contributions of our method.
>   Regarding the spatial-frequency enhancement module (i.e., Region Gaze Module), we introduced an adaptive thresholding mechanism to distinguish high-frequency information from low-frequency components, unlike the fixed thresholds used in previous studies. This design advantage allows for better differentiation between noise and signal, reducing the impact of noise. Furthermore, we designed a region amplifier module that amplifies areas of interest for secondary recognition, which significantly enhances the reconstruction quality. The ablation study results confirm the notable contributions of the region amplifier to the overall system performance.

---

> > ### Comment · Reviewer_EKVY · 2024-11-25
> >
> > Thanks for your detailed response. While the authors partially addressed my concerns, the contribution of this setting is still limited since existing works have already introduced object detection into lensless imaging. Hence, I tend to keep my rating.

---

> > > ### Author Response · Authors · 2024-11-26
> > >
> > > Thank you very much for your positive feedback. In your response, you mentioned: " *Thanks for your detailed response. While the authors partially addressed my concerns, the contribution of this setting is still limited since existing works have already introduced object detection into lensless imaging. Hence, I tend to keep my rating.* "
> > >
> > > We would like to emphasize that **the core contributions of our work do not lie merely in the  network modules but rather in a comprehensive and thoughtful integration of contributions across tasks, datasets, engineering applications, and methodological considerations.** We sincerely hope the reviewer can recognize the effort and depth of contribution we have invested in this work.
> > >
> > > To further clarify this point, we provide a more detailed explanation of the key contributions underlying our study.
> > >
> > > **1.Task Contribution**
> > >
> > > We would like to emphasize that our work is the first to address camouflaged object detection (COD) in the context of lensless imaging, focusing on the unique challenges associated with high-level reasoning tasks in this domain. This research provides a novel technical pathway for advancing the practical application of lensless imaging technologies.
> > >
> > > It is important to acknowledge that some existing studies have indeed explored object reasoning tasks in lensless imaging [1][2][3]. However, these efforts are either limited to basic object classification [1] or conventional object segmentation [2][3], which face significantly fewer challenges compared to our lensless COD framework. Our method represents a paradigm shift, offering a comprehensive solution to enhance lensless object reasoning under complex conditions. **This advancement holds considerable value for addressing intricate object reasoning problems in real-world environments and provides key insights for extending lensless imaging technologies to practical applications such as surveillance, reconnaissance, vivo diagnostics, and IoT systems. **
> > >
> > > Moreover, as demonstrated by the comparative results, our method significantly outperforms existing methods, including state-of-the-art COD methods, in terms of task performance (e.g.,  **Tab. 1, Fig. 4, Fig. 5, Tab. 5, Fig. 10, and Fig. 11** ). **We sincerely hope the reviewer will recognize our efforts in enhancing the practicality of computational imaging technologies and advancing task-level contributions in computer vision, and provide a more comprehensive evaluation of our  contributions in this new task domain.**
> > >
> > > **2. Dataset Contribution**
> > >
> > > We present the first dataset for lensless COD, named the DLCOD dataset. This dataset establishes a crucial benchmark for evaluating lensless imaging systems in the context of COD tasks, providing valuable insights for expanding both the lensless imaging and COD research communities. **We hope the reviewer recognizes our substantial efforts in broadening the scope of lensless imaging (and by extension,**** emerging** ** computational imaging techniques) within its application domain. We respectfully request the reviewer to reconsider the dataset's contribution and its significant contribution to the field** .
> > >
> > > **3. Engineering Application Contribution**
> > >
> > > Our study represents a pioneering attempt to engineer practical applications of lensless imaging technology, offering valuable perspectives on **its deployment in areas such as surveillance, reconnaissance, ****vivo**** diagnostics, IoT systems, and confined-space detection.** Specifically, lensless imaging systems bring unique advantages, including reduced size, cost, and power consumption, alongside unparalleled flexibility in design tailored to specific use cases—features unattainable by conventional imaging devices. Furthermore, lensless systems inherently support privacy protection, a critical limitation in traditional imaging systems. By integrating lensless imaging devices with task-specific reasoning algorithms, we expand the practical boundaries of inference applications, addressing the demands of complex engineering scenarios. **We sincerely hope the reviewer acknowledges our efforts to advance the engineering applicability of lensless imaging technologies and appreciates the latent value of our work in addressing real-world challenges.**

---

> ### Author Response · Authors · 2024-11-25
>
> **Q2：Insufficient experiments.**
>
> We value your suggestions for improving experimental comparisons and have addressed your concerns below:
>
> - **SOTA COD Comparisons:** Figs. 4-5 and Table 1 in the original manuscript already compare multiple SOTA COD methods. As requested, we have added experiments using methods [3] and [4] in the revised manuscript (in the **Appendix A.3, Fig. 10, and Tab. 5**). Unfortunately, [2] lacks publicly available code, preventing direct comparisons. However, we have included a detailed discussion of its methodology to contextualize our findings. Here, **in following Tabs. 1 and 2**, we conduct a brief experiment to demonstrate the performance. The results indicate that both methods remain inferior to our proposed method.
> - **Two-Stage Methods:** We have conducted comparative experiments with two-stage methods, as detailed in **Appendix A.2 (including Fig. 9 and Tab. 4). **The results show that the complexity of two-stage methods (methods with “detection-after-reconstruction strategy”) has increased severalfold, which is evidently detrimental to practical engineering applications, in terms of FLOPs and parameters. Our one-step method achieves performance metrics within 10% of the best results in some aspects, while significantly reducing computational complexity, making it more advantageous for practical applications. Furtheromre our one-step method enhances privacy protection by eliminating visual information from the process, thereby extending its applicability to real-world scenarios.
>
> **Table 1** Comparison result on method listed in comments(FPNet, FSPNet) on Test-Easy dataset
> | Method |  $F_{\beta}^{\omega}$|  $M$ |  $E_{\xi}$ |  $S_{\alpha}$ |
> |--------|-------------------------------------|---------------------|---------------------------|-----------------------------|
> | FPNet  | 0.741                               | 0.113               | 0.837                     | 0.797                       |
> | FEDER  | 0.757                               | 0.105               | 0.845                     | 0.816                       |
> | Ours   | 0.815                               | 0.079               | 0.896                     | 0.834                       |
> |        |
>
> **Table 2** Comparison result on method listed in comments(FPNet, FSPNet) on Test-Hard dataset
> | Method |  $F_{\beta}^{\omega}$|  $M$| $E_{\xi}$ |  $S_{\alpha}$ |
> |--------|-------------------------------------|---------------------|---------------------------|-----------------------------|
> | FPNet  | 0.539                               | 0.159               | 0.746                     | 0.712                       |
> | FEDER  | 0.584                               | 0.131               | 0.757                     | 0.724                       |
> | Ours   | 0.705                               | 0.098               | 0.845                     | 0.770                       |
> |    |
>
>
> Reference:
>
> [1] Khan Salman, Siddique, Sundar Varun, Boominathan Vivek, Veeraraghavan Ashok, and Mitra Kaushik. Flatnet: Towards photorealistic scene reconstruction from lensless measurements. IEEE Transactions on Pattern Analysis and Machine Intelligence, 44(4):1934-1948, 2022.
>
> **Q3: Details of the proposed dataset.**
>
> The real dataset utilized in this study and the dataset described in [5] are both derived from the same source: a subset of ImageNet comprising 10k data pairs. In this work, the dataset was curated specifically for COD task requirements, resulting in a refined subset of 2.6k data pairs. Conversely, the dataset in [5] was curated for general object segmentation with broader selection criteria, leading to a larger subset of 5.9k data pairs. While both datasets are sampled from the same original source, some overlap is inevitable. Based on our analysis, the overlap consists of 326 data pairs. We have provided a detailed analysis of these differences in the **Sec 4.1** of revised manuscript to ensure clarity and to highlight the unique focus of our dataset for COD.

---

> > ### Author Response · Authors · 2024-11-25
> >
> > **Questions:**
> >
> > **Q1:Does the performance improvement come from the large parameters and computational cost?.**
> >
> > While larger models typically outperform smaller ones, this is not always guaranteed. As shown in our comparative results (Table 1 in manuscript), EyeCoD, with the highest FLOPs, and OCENet, with the largest number of parameters, both exhibit significantly worse performance than our proposed method.
> >
> > **Q2:Error in Figure 2, where the sigmoid function is directly fed to addition without any input in PVT.**
> >
> > Thank you for pointing out the error in Fig. 2. We have corrected it in the revised version of the figure and updated the manuscript accordingly. The figure now accurately represents the intended architecture, where the sigmoid function is correctly provided with the necessary input.

---

> ### Author Response · Authors · 2024-11-26
>
> **4. Methodological Contributions:**
>
> While the reviewer expressed concerns about the novelty of our method, we would like to clarify that our method is not a aggregation of existing works but a carefully coupled design inspired by the mechanisms of "confirmation (fixation) and localized focus (magnification)" employed by human vision to observe uncertain or camouflaged objects.
>
> For the **OFE** module, while it may seem similar to designs in [2][4], we emphasize that our OFE essentially implements Wiener filtering, a principle widely adopted in various works [5][6]. As stated in our introduction, the uniqueness of our OFE lies in its task-driven learning of spatial information (e.g., camouflaged object regions), which differentiates it from the reconstruction-oriented designs in [2][4].
>
> Regarding the **Spatial-Frequency Enhancement**, our method is rooted in the types of information required for human object recognition, integrating both spatial and frequency domains. Unlike the fixed frequency decomposition in existing methods, our adaptive frequency selection mechanism introduces a novel adaptive thresholding strategy to decouple high- and low-frequency components. This adaptability is crucial for noise-invariant information separation and improved inference performance, an aspect overlooked in prior research. As evidenced by our comparative experiments ( **Fig. 10, Fig. 11, and Tab. 5** ), our method demonstrates clear advantages. **We hope the reviewer to recognize the distinctions in our mechanism design compared to existing works.**
>
> Additionally, we propose a **Region Amplifier** module, which does not merely magnify the scene but leverages the coarse localization mask provided by the first RGM to adaptively enhance the object region. This selective amplification enriches the fine details of the region of interest, significantly improving recognition quality in the subsequent stages. **Importantly, the design is intentionally lightweight, involving only a single convolution layer and simple mathematical operations, ensuring minimal computational complexity. **
>
> In summary, our work is not an aggregation of network elements but represents holistic innovation across tasks, datasets, engineering applications, and methodological paradigms. We sincerely encourage the reviewer to evaluate our work comprehensively rather than limiting the focus to network design aspects. We firmly believe our contributions offer significant value to the relevant research domains, and we respectfully reaffirm our request for a thorough assessment of our work.
>
> Reference:
>
> [1] Xiuxi Pan, Xiao Chen, Tomoya Nakamura, and MasahiroYamaguchi. Incoherent reconstruction free object recognition with mask-based lensless optics and the transformer. Optics Express, 29 (23):37962-37978, 2021
>
> [2] Xiangjun Yin, Huanjing Yue, Mengxi Zhang, Huihui Yue, Xingyu Cui, and Jingyu Yang. Inferrin objects from lensless imaging measurements. IEEE Transactions on Computational Imaging, 8: 1265-1276, 2022.
>
> [3] Haoran You, Yang Zhao, Cheng Wan, Zhongzhi Yu, and et al. EyeCoD: Eye tracking system acceleration via flatcam-based algorithm & hardware co-design. IEEE Micro, 43(4):88-97, 2023.
>
> [4] S. Khan and V. Sundar and V. Boominathan and A. Veeraraghavan and K. Mitra, et al. FlatNet: Towards Photorealistic Scene Reconstruction from Lensless Measurements. IEEE Transactions on Pattern Analysis & Machine Intelligence, 2020.
>
> [5] Jiangxin Dong, Stefan Roth, and Bernt Schiele. Deep wiener deconvolution: wiener meets deep learning for image deblurring. In Proceedings of the 34th International Conference on Neural Information Processing Systems. 89, 1048-1059, 2020.
>
> [6] Dong, Jiangxin et al. DWDN: Deep Wiener Deconvolution Network for Non-Blind Image Deblurring. IEEE Transactions on Pattern Analysis and Machine Intelligence, 44, 9960-9976, 2021.

---

> > ### Comment · Reviewer_EKVY · 2024-12-03
> >
> > Given the detailed response from the authors, I have decided to update my score from 5 to 6. Still, the contribution of this work is borderline overall in my opinion.

---

### Official Review · Reviewer_pe4y · 2024-11-05

**Soundness:** 3
**Presentation:** 3
**Contribution:** 3
**Rating:** 6
**Confidence:** 4

**Summary:**

The authors introduce a new method for detecting concealed objects using a lensless camera, the Region Gaze-Amplification Network (RGANet), which progressively enhances concealed object detection through well-crafted feature extraction and amplification techniques. A novel real-capture dataset is proposed for training and evaluate the proposed method.

**Strengths:**

The paper is clearly written, and the experimental results are compelling. The proposed new real-capture dataset DLCOD will help further research in this field. Additionally, the authors have discussed the limitations of their proposed method in the appendix.

**Weaknesses:**

There are some aspects that could benefit from further clarification and enhancement:

1. Additional details about the setup of the real-capture experiments would enhance the reproducibility and understanding of the method. Specifically, could the authors provide information on the distance between the PHlatCam and the display, as well as the display's specifications (e.g., size, model, and whether it is an LCD or OLED)?

2. Although the model was trained on a real dataset, the data was captured from a display screen. Given that lensless cameras may capture a broader range of wavelengths than standard RGB cameras, will using a screen-based dataset introduce potential bias? The model may be less effective in real-world conditions where wavelengths are not limited to the three produced by RGB displays. It would be beneficial for the authors to conduct additional experiments using non-display-based scenes to validate the model's performance in more natural, unfiltered conditions (qualitative evaluation is not required). If this is not feasible, further discussion of this limitation could be included.

**Questions:**

Please refer to the *Weakness* section.

---

> ### Author Response · Authors · 2024-11-25
>
> Thank you sincerely for these valuable comments and acknowledgment of our work. To ensure clarity and address your concerns comprehensively, we respond to each comment individually:
>
> **Weaknesses:**
>
> **Q1:Additional details about the setup of the real-capture experiments would enhance the reproducibility and understanding of the method.**
>
> We sincerely appreciate your valuable feedback regarding the setup. Below are the requested details, which we will include in the revised manuscript ( **Appendix A.1** ) as:
> **(1) Distance between the PHlatCam and the display:** The PHlatCam was positioned **42 cm** from the display throughout all real-capture experiments. This distance was carefully selected to optimize image capture, considering the camera’s field of view and resolution. This configuration remained consistent during both training and testing phases, ensuring uniform alignment of camera and monitor pixels.
>
> -**(2) Display specifications:**
>
> - **Model and Type:** The display used was a  **Dell S2425HS** , which is an **LCD** screen.
> - **Size:** The screen size was  **24 inches** , with a resolution of  **1920×1080 pixels** .
>
> **(3) Additional Notes:** The image was resized via bicubic interpolation to fit the largest central square on the monitor. The white balance for PHlatCam was calibrated using the automatic white balance setting of the PointGrey Flea3 camera, determined when an all-white image was displayed on the monitor. The exposure time was governed by the camera's automatic mode, with gain fixed at 0 dB.
>
> **Q2：additional experiments using non-display-based scenes to validate the model's performance.**
> We appreciate your concern regarding the potential bias introduced by using a screen-based dataset, especially since lensless cameras can capture a broader range of wavelengths compared to standard RGB displays.
>
> To address this, we conduct additional experiments using a dataset that captures camouflage scenarios in natural environments, free from screen-based biases. This dataset, consisting of 30 pairs of lensless imaging data, reconstructed scenes, and ground truths, covers a broader range of wavelengths, allowing us to evaluate the model performance in unfiltered, real-world conditions. The results highlight the model’s effectiveness and robustness in real-world conditions, extending its capabilities beyond screen-based data.
>
> These results are included in the **Appendix A.4 and Fig.11** of revised manuscript. However, we acknowledge that a larger and more diverse set of non-display-based data would offer even more robust validation of the model's performance. This will be explored in future work to further assess and refine the model across a wider range of real-world scenarios.

---

### Meta-Review · Area_Chair_my2m · 2024-12-21

**Metareview:**

The paper introduces the Region Gaze-Amplification Network (RGANet), a novel method for detecting concealed objects using a lensless camera. The method employs a progressive approach to enhance concealed object detection through advanced feature extraction and amplification techniques. Additionally, the authors propose a new real-capture dataset tailored for concealed object detection (COD) with lensless imaging systems, providing a valuable resource for training and evaluation.

The reviewers praised the paper for its clear presentation and reasonable experimental results. The introduction of a real-capture dataset specifically designed for COD is also expected to benefit the broader research community.

 Based on the reviewers' unanimous recommendations and the authors' successful rebuttal, I recommend accepting this paper.

**Additional Comments On Reviewer Discussion:**

The authors' responses during the rebuttal addresses part of the reviewers' concerns, like the experimental setup and generalisability of the method. Finally, reviewers  unanimously recommend for acceptance.

---

### Decision · Program_Chairs · 2025-01-22

Accept (Poster)